# A Note on Representational Understanding

**DOI:** 10.3390/e24091313

**Published:** 2022-09-17

**Authors:** Antal Jakovác, András Telcs

**Affiliations:** 1Department of Computational Sciences, Wigner Research Centre for Physics, H-1121 Budapest, Hungary; 2Department of Computer Science and Information Theory, Faculty of Electrical Engineering and Informatics, Budapest University of Technology and Economics, H-1111 Budapest, Hungary; 3Department of Quantitative Methods, Faculty of Business and Economics, University of Pannonia, H-8200 Veszprém, Hungary

**Keywords:** data representation, coordinate systems, entropy, first and second error

## Abstract

In this paper, we explore a new approach in which understanding is interpreted as a set representation. We prove that understanding/representation, finding the appropriate coordination of data, is equivalent to finding the minimum of the representational entropy. For the control of the search for the correct representation, we propose a loss function as a combination of the representational entropy, type one and type two errors. Computational complexity estimates are presented for the process of understanding and using the representation found.

## 1. Introduction

Intelligence in general is the ability to respond quickly and adequately to challenges of the external world. Animals need to be able to recognise enemies and predators quickly. This can be considered as a one-sided classification, one class versus all the others. In [1] the authors argue that responding extremely quickly requires a different solution than classical classification or learning. There, they propose a new paradigm, fundamentally different from classification, to cover the cognitive process of understanding. The present paper shows the theoretical feasibility of the comprehension process and proposes practical development of the  process of understanding.

The authors in [1] argue, following the strategy of representation learning [2,3], that understanding a topic is equivalent to finding the right representation of the data. According to this approach, understanding does not involve data compression, but merely the rearrangement of known facts/characteristics that fit best to the observed phenomena.

There is a subtle philosophical difference between understanding and learning, although they may seem technically similar and are part of our cognitive work. Classic AI/ML tasks such as classification, clustering, encoding, etc. go hand in hand with understanding, “model building”. Before we are able build a proper model, we try to classify objects based on their characteristics, features, or try to cluster them. If we find a strikingly good arrangement, it can lead to a model idea. After some trials we find a model that fits the data well (after adjusting a few parameters) and meets our expectations of the model. We stress here that model building always involves some preliminary assumptions (which can be good or bad). Once you have a suitable model in hand, the task of classification or clustering is easy. As an example, let us consider a classification problem. If we have to classify a new item, we have to compare its features with those of the possible classes. If we have a model, a few relations between some features should be checked to identify the right class. This is the bird tweet phenomenon. Without a model, it is difficult (expensive in terms of algorithmic steps) to find the solution, while the model plays the role of the bird and tells us the correct answer. Imagine that we have sample points from several polynomials of order one, two and three. If we are given a new set of sample points from an unknown polynomial, even if we know the three classes, we need to compare several class elements with the new one until we find a class element that is convincingly close, similar, to the new one. Contrary, if we know the laws of the three classes, a quick model fitting yields the answer. We will see that this quick solution comes at a price. While supervised or unsupervised learning algorithms have relatively low complexity, finding a model that well characterizes the classes, the relationship between the features, is a very difficult task with much higher complexity. The difference is typically exponential, or even hyper-exponential, depending on the task.

Let us take a closer look at the learning and model building processes. We talk about training when we have a feature set of data and we want to fine-tune a combination of these features (pick out a few key ones) to optimise some behaviour (usually classification or clustering, in some cases forecasting or filling in missing data). The task of understanding, on the other hand, is to find the best data model that reveals the relationships (functions, laws) between features and best describes a problem that is set prior the whole investigation. We call this the context of model building. Training always requires an explicit loss function that decides whether the trained system behaves correctly or not. This may be rather implicit, as in the case of auto-encoders or reinforcement learning. In the process of understanding, we consider only the input data set and try to figure out the feature functions that separate our data set from the rest of the world.

More formally, in the task of understanding the representative elements of the system (sub-universe) we want to understand, are presented one by one to the model-building apparatus. It has separating and descriptive coordinates (paper [1] called them relevant and irrelevant coordinates, respectively, based on observed features). These are either kept as they are or modified to better fit the new observed element. The separation coordinates are expected to be constant 1 for the elements of the investigated sub-universe. Meanwhile, the descriptive coordinates distinguish the elements of our subset piece-by-piece. After observing a large enough number of samples, we expect to have a good understanding, and later our understanding can also be the basis for a quick classification.

Again, we emphasize that we think that there is no understanding/model building without context, prior assumptions, questions and prior elimination of almost all the aspects of the universe, except some that are the particular subject of the investigation, the problem to be solved. In problem setting, we must specify the set of objects Ω and the function space that maps the “measurements” of the elements of the space to the new “coordinates”. In our abstract model, the sub-space of binary functions which has $log_{2}\left(\Omega \right)$ variables should be specified. Overall, the context is given by Ω and the chosen function space.

One simplified example could be the predefined scale of the study. We may search for a model of sand in a desert. If we have bird’s eye view photographs of the desert ($10^{6}$ mm scale), we can try to shift and overlap the dunes in the images and find that there is an almost periodic pattern that describes well the sand surface. The model can be formalised using trigonometric series. Time lapsed video recordings of the same scale can reveal wave-like behaviour and explain the periodicity (or falsify the static view). At a $10^{1}$–$10^{2}$ mm scale, surface tension, friction, avalanche effects may contribute to the description of the observation. The formalism should be based on polynomials of a few order, chaotic systems, or stochastic dynamical systems. On a scale of $10^{-3}$–$10^{-0}$ mm, fragmentation, particle collision processes can be captured and the formalism can be based on branching processes or dynamical systems with external potentials (forces) from statistical physics.

In general, it is not easy to decide whether a coordination is appropriate in the above sense or not. Are we satisfied with the coordination (representation) or do we need further investigation, sample elements? To facilitate this, it was suggested in [4] to associate an entropy with each representation, which is minimal if the coordinates are chosen properly. In [4], some properties of the proposed entropy function were investigated and it was demonstrated in simple examples that it indeed performs the desired task.

In the present paper, we take a closer look at the practical understanding process. First, we recall the formal model of understanding. We then show that representation entropy is minimal if and only if the representation is canonical, i.e., a representation that separates the subset and describes the elements in terms of “independent” coordinates. We propose a loss function for representations and provide theoretical and practical calculation of the type one and type two errors of representations. Finally, an estimate of the cost/complexity of the understanding procedure is given.

## 2. The Representational Entropy

We assume that Ω is a finite set,$\left\{\Omega ,F,P\right\}$ is a probability space, and *P* is uniform over Ω: $P\left(\omega \right)=\frac{1}{\left|\Omega \right|}$ for ω∈Ω.

For simplicity, we use sets with cardinality $2^{k}$ with k>0 integers. $\left|\Omega \right|=2^{N}$. Let $\Omega_{1}\subset \Omega$ with $\left|\Omega_{1}\right|=2^{N_{1}}$.

**Definition** **1.**
*A coordination of *Ω* is a bijection $x={\left\{x_{i}\right\}}_{i=1}^{N}:\Omega \to {\left\{0,1\right\}}^{N}$. The coordinates are the binary functions: $x_{i}:\Omega \to \left\{0,1\right\}$ for i=1,…,N.*


For continuous variables see the discussion in [1].

**Definition** **2.**
*A representation of $\Omega_{1}\subset \Omega$ is a coordination in which the set $\Omega_{1}$ and its complement $\Omega_{2}=\Omega \backslash \Omega_{1}$ are distinguished by $N-I_{1}$ coordinates (binary functions), called separation coordinates (SCs), if all of them are 1 on $\Omega_{1}$, and at least one of them is zero on $\Omega_{2}$. The other $I_{1}$ coordinates called Descriptive Coordinates (DCs). Given that permutation of coordinates is irrelevant we may assume that the separation coordinates have the lower index:*

$$\begin{array}{cccc} \forall i\in \{1,\ldots ,N-I_{1}\}\; & x_{i}\left(\omega \right)=1 & \mathrm{if} & \omega \in \Omega_{1},\\ \exists i\in \{1,\ldots ,N-I_{1}\}\; & x_{i}\left(\omega \right)=0 & \mathrm{if} & \omega \notin \Omega_{1}. \end{array}$$



**Definition** **3.**
*A representation of $\Omega_{1}$ is optimal (canonical) if all the DCs as random variables are independent over $\Omega_{1}$ (with respect of the uniform distribution over *Ω*).*


Let us denote the descriptive coordinates by $y={\left\{y_{i}\right\}}_{i=1}^{I_{1}}$, and the function they provide by $y:\Omega_{1}\to {\left\{0,1\right\}}^{I_{1}}$ as well.

It is shown in [1] that the optimal (canonical) coordination always exists and coincides with the binary labeling of the elements of $\Omega_{1}$ by descriptive coordinates, that is, by labeling them with the $N_{1}$-length binary vectors. One should observe immediately that $y:\Omega_{1}\to {\left\{0,1\right\}}^{N_{1}}$ is a bijection for the canonical representation.

**Definition** **4.**
*The representation entropy is defined as follows: let $p_{i}\left(\sigma \right)=P_{\Omega_{1}}\left(x_{i}\left(\omega_{i}\right)=\sigma \right)$, $\sigma \in \left\{0,1\right\}$:*

$$S_{rep}\left(x,\Omega_{1}\right)=-\sum_{i=1}^{N}\sum_{\sigma =0}^{1}p_{{}_{i}}\left(\sigma \right)\log_{2}p_{{}_{i}}\left(\sigma \right).$$



For convenience, let $H\left(x\right)=S_{Shannon}\left(x\right)$ denote the classical Shannon entropy:$$H\left(\xi \right)=\sum_{k}-p_{k}\log\left(p_{k}\right),$$
where $p_{k}=P\left(\xi =x_{k}\right)$. Using that notation we have the following observation.

**Lemma** **1.**
*The representation has minimal representational entropy if and only if the coordinates are independent (or in other words, the representation is complete).*


**Proof.** At first, let us note that the entropy $H(y|\Omega_{1})$ of the DCs is constant, equal to $N_{1}$, which justifies the search for a minimum. The statement of the Lemma is just reformulation of the well-known fact that
(1)$$H\left(z_{1},z_{2},\ldots ,z_{n}\right)\leq \sum_{i=1}^{n}H\left(z_{i}\right)$$
and equality holds if and only if all the $z_{i}$-s are independent. We give a short proof of this statement since it is in several textbooks (e.g., [5] ) but not all the proofs are clear or complete.If the family of random variables $z={\left\{z_{i}\right\}}_{i=1}^{n}$ is independent then we have the equality in (Equation 1) by the chain rule. Let us assume that the family of random variables *z* is not independent. In that case, there are $I_{1}$ and $I_{2}\subset \left\{1,2,\ldots ,n\right\}$ such that $I_{1}\cap I_{2}=\emptyset$, $I_{1}\cup I_{2}\subset \left\{1,2,\ldots ,n\right\}$, $I_{1},I_{2}\neq \emptyset$ and $x={\left\{z_{i}\right\}}_{i\in I_{1}},y={\left\{z_{j}\right\}}_{j\in I_{2}}$ such that the variables *x* and *y* are not independent and
$$H\left(z\right)=H\left(x,y\right)≨H\left(x\right)+H\left(y\right).$$For convenience let us reorder $z_{i}-s$ so that $I_{1}=\left\{1,..k\right\},I_{2}=\left\{k+1,\ldots ,n\right\}$. Using that notation we have that
$$\begin{array}{ccc} H\left(z_{1,}z_{2},\ldots ,z_{n}\right) & = & H\left(z\right)≨H\left(x\right)+H\left(y\right)\leq \sum_{i=1}^{k}H\left(z_{i}\right)+\sum_{i=k+1}^{n}H\left(z_{i}\right)\\ H\left(z_{1,}z_{2},\ldots ,z_{n}\right) & ≨ & \sum_{i=1}^{n}H\left(z_{i}\right), \end{array}$$
which shows the reverse implication. □

## 3. Search

Let us assume that we have a given coordination *x*. Without loss of generality, we can assume that the first $N-I_{1}$ coordinates are going to represent $\Omega_{1}$ with value 1-s. The type one error is:$$\alpha =\alpha \left(x,\Omega_{1}\right)=P\left(\cup {\left\{x_{i}\left(\omega \right)=0\right\}}_{i=1}^{N-N_{1}}|\omega \in \Omega_{1}\right)$$
and the type two error is:$$\beta =\beta \left(x,\Omega_{1}\right)=P\left({\left\{x_{i}\left(\omega \right)=1\right\}}_{i=1}^{N-I_{1}}|\omega \notin \Omega_{1}\right).$$

The loss function is then defined as
$$L\left(x,\Omega_{1}\right)=S_{rep}\left(x,\Omega_{1}\right)+\lambda \alpha \left(x,\Omega_{1}\right)+\mu \beta \left(x,\Omega_{1}\right)$$
where λ,μ≥0 regularizing meta parameters.

One should take into consideration when choosing the meta-parameters λ,μ that
$$\begin{array}{ccc} \log_{2}\left|\Omega_{1}\right| & \leq & S_{rep}\left(x,\Omega_{1}\right)\leq N,\\ 0 & \leq & \alpha ,\beta \leq 1. \end{array}$$

The theoretical values of the error probabilities are cumbersome, but interesting on their own. Their detailed calculation is given in Appendix A. The empirical estimates are easy. Similarly to the ML procedure, we split the sample into two. We use the first to create a coordination of the learning set, assuming that we have large enough sample then with the second part we calculate the estimate of the type one error. If we have access to non $\Omega_{1}$ elements, then we can calculate the estimate of the probability of type two error as well.

### Costs

The representation cost can also be a factor in the design of the coordination. There are different costs.

1.Creation of the coordinates that include the presentation of ω-s, building of the $x_{i}$ coordinates and, at least once, the calculation of the ready coordinates.2.Storage of the coordinate functions.3.Calculation of the representation entropy, type one and type two errors.4.Cost of decision if $\omega \in \Omega_{1}$ and full coordination of ω.

Let $\Omega_{2}\subset \Omega_{1}$ be the set of presented items. The creation of a complete coordination of $\Omega_{2}$ needs the presentation of $\left|\Omega_{2}\right|=2^{N_{2}}$ elements. The canonical, complete representation contains $r_{2}=N-N_{2}$ functions $x_{i}{|}_{\Omega_{2}}\equiv 1$ and further $N_{2}$ functions which maps to the lexicographical ordered binary vectors of length $N_{2}$. All the functions should be different. A very conservative estimate of the construction cost is the typical length $l\left(N\right)$ of a binary function of *N* inputs: (c.f. [6])
$$l\left(x_{i}\right)=l\left(N\right)∼2^{N}/\log_{2}N$$Of course if the function class is restricted, much lower cost is possible, but without the explicit knowledge of that we can not incorporate into our estimate. We need *N* such functions (where the choice of the value of *N* is based on expert guess, suggestion, external parameter of the process, can not be specified by the procedure). The cost of learning $C_{L}$ is estimated by
$$\begin{array}{ccc} C_{L} & = & \mathrm{number}\;\mathrm{of}\;\mathrm{items}\times \mathrm{number}\;\mathrm{of}\;\mathrm{functions}\;\times \cos t\;\mathrm{of}\;a\;\mathrm{function}\\ & & ∼c2^{N_{2}}N2^{N}/\log_{2}N, \end{array}$$
where c≥1 given that each constructed function should be calculated at least once for each $\omega \in \Omega_{2}$.

The storage and recall of the representation needs at least
$$C_{L}∼c^{\prime }N2^{N}/\log_{2}N$$
steps.

The representation entropy for a complete representation is fixed ( $1/M_{2}$ ), no calculation needed. Type one error can be estimated with the observed relative frequency which is based on the investigation of $M_{3}$ number of $\Omega_{1}$ elements. That needs
$$\begin{array}{ccc} C_{test} & = & \mathrm{number}\;\mathrm{of}\;\mathrm{items}\times \mathrm{number}\;\mathrm{of}\;\mathrm{functions}\;\times \cos t\;\mathrm{of}\;a\;\mathrm{calculation}\;\mathrm{of}\;a\;\mathrm{function}\\ & ∼ & c^{\prime }M_{3}\left(N-N_{2}\right)2^{N}/\log_{2}N \end{array}$$
steps. If we have access to, or can create, $\Omega \backslash \Omega_{1}$ elements for test purposes, a similar estimate can be given for the cost of the estimate of type two error.

Finally, from the above consideration, it is clear that for an ω the decision if $\omega \in \Omega_{1}$ needs
$$C_{decision}∼\left(N-N_{2}\right)2^{N}/\log_{2}N$$
steps and the ull representation needs
$$C_{full}∼N2^{N}/\log_{2}N$$
steps.

## 4. Summary and Discussion

The goal of understanding is to build a data model by which we can make a distinction between the “essential” an “unimportant” items.

Following [1], we identify the set of essential data through their characteristic features. This means that we calculate some quantities that have a given value for our distinguished set. We should build our data representation based on the characteristic and irrelevant features, which are called separation and descriptive coordinates in this paper. Once we know all the separation coordinates, we can identify our singled out set with 100% certainty.

In practice, however, it is very tedious to find this coordination and verify if it is optimal. It is much easier if we have a real valued function (loss function) that is minimal for the correct data representation. In this work, we have proven with mathematical rigour that the representation entropy defined in Definition 4 has exactly this property.

It is worth to emphasize that this representation quality measure is of statistical nature. While the generally used loss functions can be computed for each input, the use of representation entropy requires the knowledge of the bitwise probability distribution of the outputs, and thus it is available only after having seen enough examples from the given set. Once we know the bitwise probabilities, we can compute the representation entropy with the psuedo-code given by Algorithm 1.
**Algorithm 1** Representation Entropy from bitwise probability distributions1:**procedure***S*($p\in {\left[0,1\right]}^{N}$)              ▹*p* are the bitwise probabilities of 1.2:    S←03:    **for** i=0,1,⋯N−1
**do**4:        $S\leftarrow S-p_{i}\log_{2}p_{i}-\left(1-p_{i}\right)\log_{2}\left(1-p_{i}\right)$5:    **end for**6:    **return**
*S*7:**end procedure**

In this paper, we worked in an abstract world with complete information, meaning that we know all possible inputs, and we can approach all elements of our specific set. In this case, we can find the best data representation using the psuedo-code of Algorithm 2, based on evolution algorithm.
**Algorithm 2** Evolution Algorithm to find the proper representation**Require:** N← number of bits of the input Ω1← set to be represented**Ensure:** initialization R0←X→{0,1}N original coordination (e.g., pixels) Sbest←N
 1:**for** a given number of iteration **do** 2:    P← random {0,1}N→{0,1}N bijection (permutation) 3:    x←(0,0,⋯0)
*N* dimensional zero vector 4:    **for** ω**in**Ω1
**do** 5:        x←x+P(R0(ω)). 6:    **end for** 7:    p←x/|Ω1|            ▹ These are the bitwise probabilities of 1. 8:    **if** S(p)<Sbest
**then** 9:        Pbest←P10:    **end if**11:**end for**12:**return**Pbest∘R0

This algorithm works because the minimal representation entropy corresponds to the best coordination of the data.

In fact, in the abstract world we can also construct the best representation (c.f. [1]) as follows. The original $R_{0}$ representation associates an integer number to all elements, interpreting its binary coordinates as digits of a binary number. Here the $\Omega_{1}$ set usually shows up randomly. However, we can find a permutation *P* that sends the element of $\Omega_{1}$ to the top of the list. Then $P\circ R_{0}$ associates the numbers $0,1,\cdots 2^{N_{1}}-1$ with the elements of $\Omega_{1}$, i.e., only its first $N_{1}$ bits can be different from one. In this representation, therefore, the first $N_{1}$ bits are uniformly distributed, the higher bits are all one, thus the representation entropy is $N_{1}=S_{Shannon}$. The type one and type two errors are all zero. Therefore $P_{best}=P$.

This solution, however, is not feasible in practice, since we do not have complete information. Then we shall use the given pseudo-code, with two modifications. The first is that we can not sample the whole ${\left\{0,1\right\}}^{N}\to {\left\{0,1\right\}}^{N}$ bijections, since the cardinality of this space is $2^{N}!$, which is usually too big to be parametrized. In practice, therefore, we shall be content with a relatively small subset of all the representations, which we should choose very carefully. For an appropriate choice of the function class, we can take into account a number of arguments, we shall examine the symmetry of the system, we may consider simple function classes (such as linear or low order polynomials), and we shall rely on the field knowledge, accumulated common wisdom in the given area.

The other reason is that usually we do not know the complete Ω and $\Omega_{1}$ sets, we only see some samples of it. In fact we do not even know the cardinality of $\Omega_{1}$ in general. To mitigate the problem of the limited number of observations in $\Omega_{1}$, we may apply methods that potentially improve the convergence. In this paper we have suggested to incorporate into the cost function the type one and type two errors. That may be useful from this point of view.

Even with the best strategy, if we do not have complete information, the fixed values of the separation coordinates determine an $\Omega^{\prime }\neq \Omega_{1}$ subset. Then a number of questions arise, such as what is the relation of $\Omega^{\prime }$ and $\Omega_{1}$, how to choose the set of available co-ordinations, how many separation and descriptive coordinates are worth to keep, and so on. These questions are the subject of our future studies.

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
