# Peer review of "A Note on Representational Understanding"

_entropy, 2022, doi:10.3390/e24091313_

Round 1

Reviewer 1 Report

> "We prove that understanding/representation, interpreted as finding the proper coordination of the data, is equivalent to the finding of the the minimum of the representational entropy."

The finding is interesting, but seems to be very marginal considering the definition of information entropy.
As the authors insist, minimum of the representational entropy might be probably related with the proper coordination of the data.
However, more rigourous justification of the argument seems to make the paper more convincing, possibly with more experiments too, not just a proof.

Reviewer 2 Report

The authors study an approach of understanding the representation of the data on the basis of the representation entropy. The work is inspiring to help understand the underlying mechanism of machine learning algorithms. This paper also analyzes the theory and algorithmic complexity. 

1. Since the work is to "take a closer look of (-> at) the practical process of understanding", it is better to add some pseudo code to illustrate the algorithm given in Section 3 and further help practitioners utilize the approach proposed in this work.

2. The work will add more value if more detailed discussions are added on how to minimize the loss function based on the representation entropy. 

3. Is there any suggestion on how to check the chain independence of DCs?

Minor comments:

line 58: observation -> observing

line 92: an -> and

Reviewer 3 Report

This is an interesting paper looking at learning and understanding.  I very much enjoyed to concept of the paper and the approach.  Having said that, I feel the paper needs to be expanded.  The middle section of the paper that goes through the mathematics is thorough but not clearly articulated, meaning, to ensure benefit and impact to a more general audience I would strongly encourage the authors to add a sentence or two to each basic part of the equations and their derivations to aid a reader's intuition about the math.  

Finally, and most importantly, the paper needs a stronger conclusion.  Currently the paper just ends with the math.  While not inappropriate to do so, I feel a more general audience will be left with many questions and unable to articulate to meaningfulness of the article.

Round 2

Reviewer 1 Report

I don't like the first paragraph added. Please consider replacing it with more formal description.

Some sentences are in passive form. Please change them to the active form.

Some proof-reading by native speakers would make the manuscript more readable.

Rather than those, I have no issues with the revised manuscript.

Author Response

The authors thank the suggestions again  We did our best to follow them.  The first paragraph is fully rewritten and thorough English revision has been made.

Sincerely

the authors

Reviewer 2 Report

My concerns have been addressed.

Author Response

We are glad to meet your expectations.

Sincerely

the authors

Reviewer 3 Report

This revision is a strong improvement.  I think the authors did a great job of addressing the concerns I stated in my original review.  The only remaining issue is simply to proof read the new material and correct a few typos and grammatical issues.  Very well done!

Author Response

(The authors gave the same response as above.)
